# A Fast Spatial Clustering Method for Sparse LiDAR Point Clouds Using GPU Programming

**DOI:** 10.3390/s20082309

**Published:** 2020-04-18

**Authors:** Yifei Tian, Wei Song, Long Chen, Yunsick Sung, Jeonghoon Kwak, Su Sun

**Affiliations:** 1School of Information Science and Technology, North China University of Technology, Beijing 100144, China; yb87403@um.edu.mo; 2Department of Computer and Information Science, University of Macau, Macau 999078, China; longchen@um.edu.mo; 3Beijing Key Lab on Urban Intelligent Traffic Control Technology, Beijing 100144, China; 4Department of Multimedia Engineering, Dongguk University, Seoul 04620, Korea; sung@dongguk.edu (Y.S.); jeonghoon@dongguk.edu (J.K.); 5Department of Computer and Information Technology, Purdue University, West Lafayette, IN 47907, USA; sun931@purdue.edu

**Keywords:** 3D spatial clustering, connected component labeling, LiDAR, GPU programming

## Abstract

Fast and accurate obstacle detection is essential for accurate perception of mobile vehicles’ environment. Because point clouds sensed by light detection and ranging (LiDAR) sensors are sparse and unstructured, traditional obstacle clustering on raw point clouds are inaccurate and time consuming. Thus, to achieve fast obstacle clustering in an unknown terrain, this paper proposes an elevation-reference connected component labeling (ER-CCL) algorithm using graphic processing unit (GPU) programing. LiDAR points are first projected onto a rasterized *x*–*z* plane so that sparse points are mapped into a series of regularly arranged small cells. Based on the height distribution of the LiDAR point, the ground cells are filtered out and a flag map is generated. Next, the ER-CCL algorithm is implemented on the label map generated from the flag map to mark individual clusters with unique labels. Finally, obstacle labeling results are inverse transformed from the *x*–*z* plane to 3D points to provide clustering results. For real-time 3D point cloud clustering, ER-CCL is accelerated by running it in parallel with the aid of GPU programming technology.

## 1. Introduction

3D obstacle perception provides a driving awareness interface for environment perception [1,2]. 3D obstacle perception is also applied in mobile obstacle recognition, obstacle tracking, remote sensing, semantic mapping, and 3D terrain reconstruction for unmanned vehicles [3,4,5]. Efficient obstacle clustering can improve the speed of traversable road recognition, surrounding obstacle avoidance, and local path planning, all of which support real-time decision making for unmanned ground vehicles (UGV) [6,7]. Traditionally, real-time obstacle clustering algorithm research involves 3D point clouds sensed by a stereo camera and video sequences mounted on UGVs [8,9]. Compared to other range sensors, light detection and ranging (LiDAR) sensors apply narrow laser beams to detect the distance to a 3D obstacle with high accuracy and speed [10]. Thus, LiDAR is widely utilized to collect 3D point clouds for fast and accurate environment perception of UGVs, particularly in obstacle clustering research for unmanned vehicles [11].

Spatial clustering, which involves partitioning 3D points into several distinguishable clusters, is the most significant process for obstacle detection during autonomous driving. However, cluster analysis of large-scale point clouds sensed by LiDAR is time-consuming. Distance and other auxiliary information are commonly utilized to evaluate the relationship between neighboring point clustering and clustering processes [12]. When computing spatial relationships between a center point and its neighboring points, the traversing neighboring points in memory incurs significant computation costs [13]. Thus, connected component detection in LiDAR point clouds is difficult to execute in real time [14,15].

Due to the unstructured and asymmetric characteristics of LiDAR point clouds, searching neighboring points is computationally expensive in the obstacle clustering process [16]. Central processing unit (CPU)-based computation methods always implement the neighboring points search process point by point, which is difficult to achieve as a real-time approach [17]. To solve these problems, a graphic processing unit (GPU)-based 3D obstacle labeling method is proposed to realize real-time obstacle clustering in LiDAR point clouds. First, all non-ground 3D points are projected onto an *x*–*z* plane and registered into a binary flag map, after ground points are filtered out by height threshold. Then, from the non-ground obstacle cells, a GPU-based elevation-reference connected component labeling (ER-CCL) algorithm is employed on the flag map to cluster connected cells into individual groups. A label map is initialized by specifying the corresponding indices of the validated cells of the flag map. In the label map generation process, the connected components are labeled by searching for the minimum index of each cell and its neighboring cells. After several iterations to update the label map, all cells in a distinguished blob are labeled with a unique value. Finally, through inverse mapping from the label map to the 3D points, the non-ground points are clustered into several individual obstacles.

Traditional obstacle clustering algorithms for LiDAR point clouds rely on a CPU to execute a computation program in a certain order [18]. The algorithm’s performance is limited by the computation speed of the CPU, i.e., CPU computation speed can be a bottleneck in achieving real time obstacle clustering of large-scale point clouds. To overcome this problem, in this study, a GPU programming method is applied to implement the iterative labeling process in parallel to speed up clustering.

The primary contributions of the proposed system are as follows: The ER-CCL algorithm with a flexible search range is suitable for processing sparse and unevenly dense LiDAR point clouds. To improve the processing speed, GPU programming technology is utilized to process the ER-CCL algorithm in parallel for each cell.To solve the problem of classifying connective obstacles, the proposed clustering method adopts height information as a reference feature in the ER-CCL algorithm to determine whether adjacent cells belong to the same obstacle.

The remaining of this paper is organized as follows. In Section 2, we discuss the studies related to obstacle clustering algorithms in 3D point clouds. In Section 3, the proposed GPU-based 3D obstacle labeling system is described. We analyze the performance of the proposed system, including clustering accuracy and processing speed in Section 4. Conclusions and suggestions for future work are presented in Section 5.

## 2. Related Works

Obstacle clustering is considered an essential preprocess for environment perception and driving awareness for UGVs [19,20]. This section surveys several 3D points clustering and obstacle labeling methods.

Traditional obstacle clustering methods in video sequences have relied on detecting foreground pixels based on the difference between two successive frames [21,22]. Arvanitidou et al. [23] realized an unsupervised moving obstacle clustering system using environmental images captured by a moving camera. Currently, deep learning and convolutional network have been applied in 2D image analysis. To provide depth information, Boulch et al. [24] exploited an RGB-Depth camera to sense 3D obstacle positions in a structural and orderly format. A 2D deep clustering network was proposed to cluster objects in a 2D view projected from the 3D points. Wei et al. [10] presented an environment perception and obstacle clustering method that used a mean-shift algorithm and a histogram map obtained from depth datasets collected by a 3D range camera. In these obstacle clustering methods, the illumination condition in an outdoor environment was not always stable, which influences the reconstruction precision of a 3D scene model. Thus, camera-based obstacle detection methods did not satisfy the precision required for autonomous driving.

Considering the limited resolution of depth and 3D range cameras, LiDAR sensing technology is suitable to capture large-scale 3D point clouds with detailed and accurate location information to realize autonomous driving for unmanned vehicles [25]. Based on the unstructured and asymmetric distribution characteristics of LiDAR point clouds, in some traditional obstacle clustering methods, a radius was defined as the spatial criterion to find neighbor points. Due to the spatial distribution characteristics, the neighbor point count of a LiDAR point in a given area was uncertain. More neighbor points existed in a dense area than in a sparse area. The uncertain number of neighboring points caused difficulty in both storage and traversing processes. To realize obstacle clustering for unmanned driving, Hackel [26] utilized a *k*-nearest neighbor (KNN) algorithm to avoid complicated iterative computation, particularly in intensive point areas, by removing duplicative points. Thus, it was possible to extract point cloud features, such as eigenvectors and eigenvalues, after reducing the redundant samples. However, unordered neighbor point searching is computationally complex; thus, the KNN algorithm was not suitable to traverse large-scale points in real-time.

To match environment perception speed with the driving speed, fast and accurate obstacle recognition and traversable road analysis was required for behavioral planning of unmanned vehicle [27]. To increase the detection speed of surrounding obstacles, Douillard et al. [16] proposed a ground and non-ground clustering method from rasterization grids on the horizontal plane. After the ground surface grids were clustered, the objects were clustered into several groups based on the adjacent relations of non-ground grids. If the height value gradients between a cell and its neighbor cells were remarkable, the occurrence rate of data edges existing at a junction point was in high probability. Then an iterative close point algorithm was utilized to extract the features of the clustered obstacles for 3D classification. Wang et al. [28] generated a hash table from an *x*–*z* plane projected from the 3D points. Given the hash table, the connected cells were clustered into several groups based on the distance between two cells. However, clustering methods that involve spatial adjacent relation analysis were not sufficient fast for 3D scanning frame computation; thus, they are difficult to apply to environment analysis of unmanned vehicle driving systems.

Clustering connected cells using CPU programming requires significant serial iterative computation, which is very time consuming, particularly for large-scale LiDAR point clouds. To increase the computational speed of the clustering process, Kalentev [29] proposed a CCL algorithm to cluster connected 2D grid cells using GPU-based parallel programming technology. In this method, the iterative and loop computations were executed in multiple GPU blocks simultaneously. With parallel computation, the time required decreased by more than 15 times, thereby achieving a fast clustering approach. To accelerate the labeling process, we employed a GPU-based CCL algorithm to cluster foreground areas in real-time surveillance videos [30]. Differing from image clustering processes, 3D LiDAR point clouds are dispersed and without structural and connected relationships among neighboring points. By projecting 3D points onto an *x*–*z* horizontal plane to establish a histogram map that shows obvious relationships among neighboring cells, we realized a GPU-based obstacle labeling method to increase the computational efficiency of connected cell clustering for obstacle clustering in the driving awareness systems for unmanned vehicles, which overcame dispersed and non-sequence issue of LiDAR point cloud.

## 3. Fast Spatial Clustering Method

A GPU-based fast spatial clustering system that separates individual clusters in LiDAR point clouds is described in this section.

### 3.1. Overview of Fast Spatial Clustering System

Figure 1 shows the flowchart of the proposed system with obstacle flag map generation and obstacle labeling functions. The input to the proposed system is a frame of the original 3D LiDAR point clouds, and the output is the spatial labeling results (as individual obstacle clusters).

In each frame, ground points generally occupy more than one-half of the original point clouds [31], and such point are not used in non-ground obstacle clustering function. Ground clustering, the first step of the proposed ER-CCL algorithm, is performed prior to executing obstacle clustering process. We propose an obstacle flag map generation method to filter out ground cells and identify non-ground cells on a 2D horizontal plane (i.e., the *x*–*z* plane).

Using the valid cells in the obstacle flag map, a label map is initialized by specifying a unique value in each label. Subsequently, we adopted ER-CCL algorithm to update every data cell’s value as a searched minimum in a defined radius. Connected valid cells belonging to the same component are labeled with the same value. After several updating iterations, each cell updates with a unique label by assigning the minimum value among its neighbors. By inversely projecting the generated label map to the corresponding 3D obstacles, each data point is assigned a unique label value, thereby achieving obstacle clustering of LiDAR point clouds.

### 3.2. Obstacle Flag Map

As the general characteristics of terrain environments, the obstacles surrounding a UGV are always perpendicular to the surface of the ground. Thus, it is feasible to implement obstacle clustering as connected component labeling on the *x–z* plane projected from 3D points. Ground cells on the *x–z* plane connect with nearly all non-ground obstacles on the ground; thus, it is difficult to separate distinguished obstacles without ground clustering.

To avoid disturbance caused by ground points, an obstacle flag map generation method is proposed to filter out the ground in a preprocess of the obstacle labeling process. The first step of the preprocess is to obtain a rough height range of ground surface by analyzing the height distribution of all points. As shown in Figure 2, the point distribution in *y*-axis exists a significant peak, which is considered as the approximate height value of ground surface. Thus, the ground height is estimated as $-\bar{h}$, where $\bar{h}$ is always considered as the LiDAR sensor height. If the height value of a point locates in a range of $-\bar{h}\pm \sigma$, this point is determined as a ground point, where the predefined variable *σ* means our allowed fluctuation range in the ground space division.

Then, all 3D points in each frame are projected onto a horizontal *x–z* plane that consists of a series of rasterized grid cells. After the 3D points are registered into their corresponding cells, a binary flag map *F* of width *u* and height *v* is updated to record the valid cells in the *x–z* plane, above which non-ground data points exist. The resolution of the flag map is denoted *s*. A point *p* (*x*, *y*, *z*) is projected onto cell *c* = (*i*, *j*), which is derived as
*i* = [*x*/*s*], *j* = [*z*/*s*],(1)

We assume that a non-ground cell contains at least one point whose height value *y_k_* locates outside the ground surface range. After all points registered on the *x–z* plane, the points projected onto a cell *c* are collected into cluster *K_c_*. Thus, a flag *f_c_* ∈ *F* is defined as
(2)$$f_{c}=\left\{\begin{array}{l} \begin{array}{cc} 1 & \underset{k\in K_{c}}{\max}(\left|y_{k}+\bar{h}\right|)\geq \sigma , \end{array}\\ \begin{array}{cc} 0 & \underset{k\in K_{c}}{\max}(\left|y_{k}+\bar{h}\right|)<\sigma , \end{array} \end{array}\right.$$

When only one ground pixel is located above a cell, $y_{k}$ + $\bar{h}$ should be less than the resolution value of the threshold value *σ*. Thus, non-ground and empty cells are filtered out. Note that binary flag map *F* is generated to record non-ground cells.

### 3.3. ER-CCL Algorithm

Based on the adjacency relation between a non-ground cell and its neighbors, the available connective cells are determined as the same data. An ER-CCL algorithm is applied to distinguish the connected components in obstacle flag map *F*.

Label map *l_c_*∈*L* is created from flag map *F*. As defined by Equation (3), label *l_c_* in *L* is initialized as its index if the corresponding *f_c_* equals to 1; otherwise, *l_c_* is set as null.
(3)$$l_{c}=\left\{\begin{array}{cc} i\times u+j & f_{c}=1,\\ null & f_{c}=0, \end{array}\right.$$

Next, the ER-CCL algorithm is applied to label the connected component from label map *L* by searching for the minimum index values of the neighbor clique of each valid cell. The neighbor clique *N_c_* of cell *c* is defined as the neighbor cell set relative to the distance from *r* to *c*. The neighbor clique is expressed as follows.
(4)$$N_{c}=\left\{\;c{}^{\prime }=\left(i{}^{\prime },j{}^{\prime }\right)||i{}^{\prime }-i|\leq r,|j{}^{\prime }-j|\leq r\right\},$$

The search range *r* is the defined size of neighbor clique. When the search range *r* is higher, the neighbor clique is larger so that the further neighbor cells are traversed. This way, excessively high or low search ranges cause over or under clustering. The distance between cells *c* = (*i*, *j*) and *c*’ = (*i’*, *j’*) is defined as
(5)$$\Delta d_{c,c{}^{\prime }}=s\sqrt{(i-i{}^{\prime }{)}^{2}+{\left(j-j{}^{\prime }\right)}^{2}},$$

The height difference sum between the highest points and the lowest points on cells *c* and *c’* is defined as
(6)$$\Delta h_{c,c{}^{\prime }}=\left|\max(y_{k})-\max(y_{k{}^{\prime }})\right|+\left|\min(y_{k})-\min(y_{k{}^{\prime }})\right|\begin{array}{cc} & k\in K_{c} \end{array},k{}^{\prime }\in K_{c{}^{\prime }},$$

Variable *E_c,c’_* is defined as the similarity value between the two cells, which is used as the evaluation criterion to constrain the minimum index searched in the label update process.
(7)$$E_{c,c{}^{\prime }}=\alpha e^{-\Delta d_{c,c{}^{\prime }}}+\left(1-\alpha \right)e^{-\Delta h_{c,c{}^{\prime }}},$$

In Equation (7), loss factor *α* is defined to balance the two influencing factors of variable *E_c,c’_,* including mapping count differences and the distance between the center cell and its singular neighbor cell. Loss factor *α* locates in (0, 1). If the count difference between a center cell and its neighbor cell is large, the similarity level between the two cells is small. In addition, the distance between the two cells is negatively correlated with their similarity level. Thus, variable *E_c,c’_* is defined to estimate the similarity level between the center cell *c* and its neighbor cell *c’*. When variable *E_c,c’_* takes a large value, the similarity level between the two cells is large. In contrast, if variable *E_c,c’_* is small, the similarity level between *c* and *c* is small. Therefore, variable *E_c,c’_* is utilized as the second constraint in the label update process.

Label map *L* is updated by specifying any valid *l_c’_* with the minimum label value among the labels of the clique *N_c_*, which is expressed as
(8)$$l_{c{}^{\prime }}=\min\left(\forall l_{c{}^{\prime }}|c{}^{\prime }\in N_{c},E_{c,c{}^{\prime }}\geq \tau \right)\begin{array}{cc} & c{}^{\prime }\in N_{c} \end{array},$$

Here, $\tau =\beta e^{-r}$. When search range *r* is increased, variable *E_c,c’_* is smaller for the cell in long distance. Thus, parameter *β* is evaluated as smaller to propagate similar cells over a long distance that are considered part of the same data.

This minimum label specifying process runs iteratively until all labels are fixed with the minimum value of their neighbor cliques. Neighbor cell *c’* is required to satisfy the condition that *E_c,c’_* is greater than or equal to similarity threshold *τ*. Finally, all labels in a connected component are labeled with the same value that is considered the unique label of the separate component. By inverse projection from the labels on the *x*–*z* plane to their corresponding 3D points, cells of the label map *L* are labeled with a unique value such that non-ground points are distinguished as several individual clusters with different labels.

### 3.4. GPU-Based Fast Spatial Clustering System

To increase the speed of the neighbor points traversing process in large-scale point clouds, GPU programming technology is applied to optimize the proposed ER-CCL algorithm to run in parallel. A GPU-based 3D obstacle labeling framework is designed by allocating point clouds, the flag map, and the label map to GPU memory with the corresponding GPU threads. Figure 3 shows the GPU functions and memory usage for obstacle labeling.

After copying raw 3D point cloud data from CPU to GPU memory, the entire process is implemented in parallel using GPU programming technology. The 3D coordinates of the sensed 3D points and their corresponding labels are created in GPU memory as the ER-CCL input and output interfaces, respectively. All 3D coordinates are projected onto the *x*–*z* plane to generate an obstacle flag map through a ground-filtering process.

The flag and label maps are also created in GPU memory to record the point existence and corresponding index in each cell. In label map, the proposed ER-CCL algorithm is executed synchronously in a series of GPU threads to search for the minimum index among each cell’s neighbors. The obstacle label map update process requires multiple iterations until the label value no longer changes. In each label update iteration, each cell in label map is allocated with a GPU thread to update its label value. When a GPU thread executes the label search process, the thread is held on to access GPU memory that stores the neighbor cells in the label maps. Based on the label map result, non-ground 3D points are inverse mapped to the corresponding label to obtain the labels of non-ground connected components to realize fast obstacle labeling.

Figure 4 shows the memory and thread allocation design in the GPU device for the proposed obstacle labeling method. The obstacle flag and label maps of same size are created and initialized in global GPU memory. In addition, point labels are also allocated in global GPU memory to store the inverse mapping result from ER-CCL clustering to identify the individual non-ground data index.

Note that the sizes of the flag and label maps are the same, and they defined by width *u* and height *v*, respectively. Each block is specified to contain *N* threads. To implement all functions synchronously, a single thread is allocated to each cell. Thus, *M=*
*u* × *v*/*N* blocks are required for parallel implementation of the functions of these maps.

The GPU-based obstacle labeling method is programmed according to the pseudocode shown as follow. The input of the ER-CCL algorithm is the LiDAR point coordinates and searching range *r*. Variables *B* and *G* are defined as the block and grid counts, respectively. The ground point filtering and obstacle flag map generation processes are implemented by the Cuda_Kernel_GroundHeightCompute() and Cuda_Kernel_FlagMapGenerate() GPU kernel programming functions. The initialization process of the proposed ER-CCL method is implemented using the Cuda_Kernel_Label_Initialization() function to initialize the label maps. The iterative label map update process is executed via a while loop with a statement to judge whether there is any label change in label_map. When the loop finished, a final label map with minimum label value is converged upon as a clustering result of rasterized cell in *x*–*z* plane. Based on the ER-CCL clustering result, an inverse mapping procedure is executed using the Cuda_Kernel_Inverse_Mapping() function.
**Algorithm 1:** GPU-based ER-CCL Algorithm**Input: point, search range *r****B*: block  *G*: grid countMemcpy (cuda point, cpu point, hosttodevice)Cuda_Kernel_GroundHeightCompute<<<*B*, *G*>>>(int height, cuda point);Cuda_Kernel_FlagMapGenerate<<<*B*, *G*>>>(cuda flag_map, cuda point, int height);Cuda_Kernel_Label_Initialization<<<*B*, *G*>>>(cuda label_map, cuda flag_map);**while** (cuda label_map is changed)  Cuda_Kernel_Label_Updating<<<*B*, *G*>>>(cuda label_maps);Cuda_Kernel_Inverse_Mapping<<<*B*, *G*>>>(cuda point_label, cuda label_map, cuda point);Memcpy(cpu point_label, cuda point_label, devicetohost);**Return: p****oint_label**

## 4. Experiments and Analysis

The obstacle clustering results obtained by our proposed system were estimated and analyzed under different parameters at different scenes in this section.

### 4.1. Dataset and Experiment Platform Introduction

In this section, the speed and accuracy of the proposed GPU-based 3D obstacle labeling method are analyzed. Experiments were performed using an EU260 unmanned vehicle produced by the BAIC Motor Corporation as shown in Figure 5. The vehicle with an HDL-32E Velodyne LiDAR sensor was driven in an outdoor environment with irregular obstacles, buildings, trees, and pedestrians. The average driving speed was approximately 30 km per hour. The proposed method was implemented on a computer with a 3.20 GHz Intel® Xeon E5-2670 CPU, a Quadro K5200 GPU, and 64 GB of RAM. The obstacle clustering results were displayed in a virtual environment programmed using DirectX software development kits.

The utilized HDL-32E LiDAR sensor generated 96,000 points in each frame. Our method was tested on more than 300 frames in different scenes, such as common roads, crossroads, squares, etc. In this project, we selected points within 20 m to test the proposed method for maintaining an accurate clustering rate in a local space. The resolution of the projected *x–z* plane was 5 × 5 cm, and the valid detection range of LiDAR point clouds was 40 m; thus, the flag map contained 800 × 800 cells.

### 4.2. Intermediate Experiment Result

Figure 6a shows the raw points in the defined valid range observed by the LiDAR sensor. After mapping all 3D points to the *x–z* plane, the flag map was generated by rasterizing continuous scanning points into *x–z* cells, as shown in Figure 6b. Figure 6c shows the connected components obtained using the proposed GPU-based ER-CCL algorithm with a specified search range of five steps. All cells in one cluster had only a single unique label value for identification, and this was utilized to generate a unique color for the cluster. Based on the clustering result in the label map, inversing mapping was executed to update the 3D point labels. Here, 3D points with same labels were considered to belong to a single data, and these 3D points were rendered in the same color (Figure 6d). To distinguish ground and non-ground obstacles, all ground points were set to light green. In addition, the data boundaries of were searched among their points, and each data was rendered with a colored bounding box (Figure 6e).

### 4.3. Time Comparison under Different Parameters

The Figure 7 illustrated the time comparison of CPU/GPU based obstacle clustering methods with different numbers of search ranges. All the time data in the figure were their average value that tested on 10 consecutive frames of a series of outdoor scenes, which were collected by our experiment platform. The blue curve in the figure means the time consumption of CPU-based obstacle clustering method. When the search range increased from 1 to 8, the tendency of average time consumption is apparently raising from 43.48 ms to 146.10 ms. Instead, the changes of our proposed GPU-based obstacle clustering methods (CCL and ER-CCL) were not conspicuous, where time consumption fluctuated around 20–40 ms with a slight increasing trend of the search ranges raising. This way, our proposed GPU-based obstacle clustering methods giving a steadier time efficiency than CPU-based clustering methods.

The clustering accuracies obtained with different numbers of search ranges were also examined, and the results are shown as clustering results in Figure 8. In Figure 8a,b, the search ranges were specified as two and four, respectively. Note that the spatial distribution of vegetation was sparse; thus, such obstacles were always clustered into several fractional parts when the search range was small. If the search range was large, individual obstacles were grouped into a single component. In our test environment, a search range of five was suitable for accurate obstacle clustering, as shown in Figure 8c. We also estimated and tested the proposed algorithm on an open dataset collected by the University of Michigan North Campus, i.e., the Long-Term Vision and Lidar Dataset (NCLT) [32]. As shown in Figure 8d–f, we used search ranges of two, four, and five to execute obstacle clustering on the NCLT dataset. The NCLT dataset contains several types of similar obstacles as our dataset, such as trees, shrubs, pedestrians, walls, and poles. In addition, the LiDAR point distributions and measurement ranges in the NCLT dataset are also similar. Considering the balance of processing speed and clustering accuracy, a search range of five was selected as the optimum value to maintain real-time and efficient obstacle clustering on both the NCLT and our dataset.

The relationships among speed, number of iterations, and cluster counts obtained for a search range of three. As can be seen, the iteration and speed curves demonstrate strong negative correlation. The labeling process of each cell was implemented in parallel; thus, the cluster counts did not obvious influence on processing speed. With the advantages of GPU parallel computing, the proposed real-time obstacle clustering method provided obstacle avoidance and traversable path detection interfaces for unmanned vehicle driving.

### 4.4. Obstacle Clustering Results under Different Scenes

This section we tested our proposed GPU-based obstacle clustering method on several different classic outdoor scenes for unmanned vehicles, including T junction (scene 1), normal road (scene 2), little square (scene 3), road with multiple trees (scene 4), road with multiple pedestrians (scene 5), and crossroad. As shown in Figure 9, the labeling results in flag maps were demonstrated in the left column, and the obstacle clustering results in real 3D scenes with corresponding bounding boxes were displayed in right column.

We analyzed the clustering details of the 6 classic outdoor scenes in Table 1, including cluster counts, iteration counts, and time consumption. All the clustering results were obtained by our proposed obstacle clustering method with search range equal to 5. Scene 1 is a T junction, where contains 4 big walls, 7 trees, 3 pedestrians, and many small trunks and pole-like objects. The total cluster counts are 301 through 50 iterations using 39.85 ms. Scene 2 display a normal narrow road with multiple walls, trees, bushes, and small pole-like obstacles. Because the scene 2 is more complex than scene 1, the iteration counts and time consumption of scene 2 are slightly higher than that of scene 1. Similar, the clustering results with corresponding bounding boxes in the other four scenes also contain multiple clear obstacles, especially these near with UGV center. Even though some obstacles were located a little further from the vehicle existed over segmentation situation, most clusters in our testing scenes were clustered correctly.

### 4.5. Obstacle Clustering Result under Different Methods

We compared the clustering accuracy of the proposed ER-CCL algorithm to those of three clustering methods, i.e., the connected component analysis algorithm [33], and the clustering in hash table method [28]. The obstacle clustering results on *x–z* horizontal planes are shown in Figure 10a,c,e. The cells belonging to one data are rendered in a distinguishing color. Based on a ground truth obstacle clustering result, the obstacles of error labeling are marked in red bounding boxes as shown in Figure 10b,d,f to analyze accuracy. Figure 10b,d,f show that several outliers and overlapping points of plants were prone to incorrect labeling and sorting. Besides, the points locating far from the LiDAR were scattered, so that it was hard to determine the clustering criteria. Thus, these sparse points were primarily divided into several small components concentrated in areas where plants were distributed densely, as shown in Figure 10d,f. 

Compared to the proposed ER-CCL algorithm, the results of the connected component analysis and clustering in hash table method showed higher error clustering frequency, especially in densely vegetated areas (top and bottom right in Figure 10d,f. Therefore, the proposed elevation-reference CCL clustering algorithm shows higher accuracy with less susceptibility to overlapping and problems related to the existence of connective obstacles.

Table 2 compares the obstacle labeling accuracy and speed performance of the three methods. Here, accuracy was estimated based on the number of clustered data and error data. Such as in Figure 10d, several individual obstacles were clustered into one data as error labeling results marked in red bounding box. The accuracy rate of the proposed method reached 98.2% with 3 false clustering obstacles among 170 valid obstacles. The data counts of error clustering were less than that using the connected component analysis, and clustering in hash table methods. In addition, the speed performance of the proposed GPU-based obstacle labeling system achieved 0.02 s each frame, was much higher than that of the other two methods.

### 4.6. Data Independency Solution of Labeling Process

Because the proposed ER-CCL algorithm is executed by GPU threads in graphic memory, the propagated label value is uncertain due to the race condition if multiple threads operate on a shared cell synchronously. To eliminate data dependencies for GPU-based labeling process, we proposed a data independency solution, which enlarge the label map to *r* × *r* intermediate label maps as shown in Figure 11a. During updating process, there was no more than two GPU threads operating one shared cell memory synchronously.

In the intermediate label maps, each cell was operated by a unique thread that independent from other threads. The (*u*/*r* + 1) × (*v*/*r* + 1) × *r* × *r* GPU threads were allocated, where u was the width of label map, *v* was the height of the label map, and *r* was searching range. During the minimum value searching process, each cell in the intermediate maps stored the minimum label value of its locating clique of the label map as illustrated in Figure 11b. Then, the label map updated with the minimum label values, which were searched from their corresponding positions in the *r* × *r* intermediate label maps. After several updating iterations, each cell in a connected component updated with a unique label by assigning the minimum value from the corresponding cells of the intermediate maps.

The data independency solution required allocating multiple intermediate label maps to eliminate data dependency phenomenon that existing in GPU-based programing. We compared the iteration numbers among the CPU-based CCL algorithm (CPUCCL) and our proposed data independence solution in GPU-based labeling process (GPUDICCL) as shown in Figure 12. Obviously, the iteration times of CPU-based algorithm were nearly less than half that of GPU-based algorithm. The main influencing factor regarding iteration differences between CPUCCL and GPUDICCL was their different mechanism on memory accessing. In CPUCCL, the left and bottom data was updated with the current data which had been updated with the minimum value of the previous updating process. However, using GPUDICCL, the minimum value searched by each thread only propagated its neighboring cells. Thus, the iteration times of GPUDICCL were more than CPUCCL algorithm. Using our proposed GPU-based ERCCL (GPUERCCL), although the minimum value also propagated to its neighboring cells, the compared cell had a certain probability to have been changed as smaller value by the other thread, due to the race condition. Therefore, GPUERCCL performed faster than GPUDICCL.

Based on our compare experiment results, the iteration times of our proposed GPU-based ER-CCL were more but the time consumption was lower than CPU-based algorithm. Even the GPUDICCL was developed as a data independence solution to eliminate the influence caused by race condition of GPU threads, more iterations were required for the convergence than GPUERCCL. Accordingly, the processing speed became low using GPUDICCL. Considering that speed performance was an important assessment to evaluate spatial clustering algorithm in point cloud domain, the proposed GPUERCCL was implemented in our obstacle detection application.

## 5. Conclusions

We have proposed a GPU-based fast spatial clustering method to label dispersed LiDAR point clouds into individual groups. In the proposed method, once ground cells are filtered by analyzing the height distribution of a height statistic map, an obstacle flag map is generated as the obstacle clustering interface. In addition, we have developed the ER-CCL algorithm to mark individual clusters with their unique labels. The search range of the ER-CCL algorithm is flexible and suitable for processing sparse and uneven density LiDAR point clouds. To achieve a real-time approach, ER-CCL was implemented using GPU programming technology to process flag and label map in parallel. The proposed method was tested on both our collected dataset and the open NCLT dataset. The experimental results demonstrate that the proposed method achieved accurate and real-time obstacle clustering in an outdoor environment. In addition, compared to other instance labeling algorithms in the sparse LiDAR point cloud processing domain, the clustering results obtained by the proposed method were obtained faster. Besides, considering the data dependence phenomenon exist in GPU-based ER-CCL algorithm, this paper gave another solution to maintain the label independency in label map updating process through allocating enough GPU memory as intermediate updating storage so that a data was only operated by one thread. In the future, we plan to extract the features of detected objects to realize an object recognition method based on our fast and accurate clustering results. The clustering results will combine with environment semantic and recognizing information to UGVs’ realize the automatic road perception, driving awareness with collision avoidance, traversal road analysis, and intelligent driving.

## Figures and Tables

**Figure 1 sensors-20-02309-f001:**
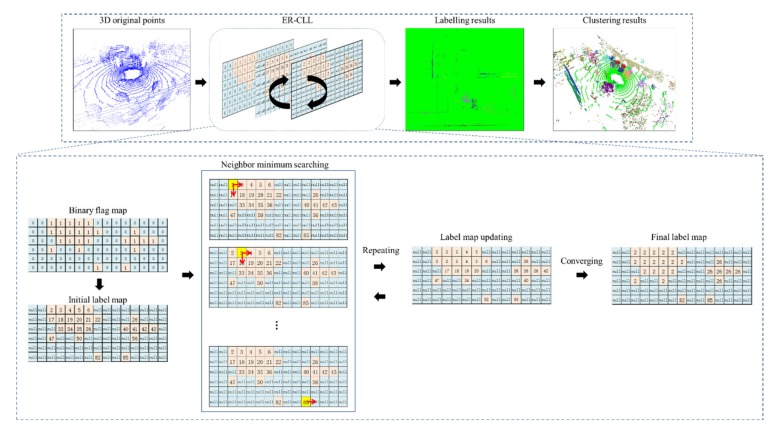
Flowchart of proposed fast spatial clustering system.

**Figure 2 sensors-20-02309-f002:**
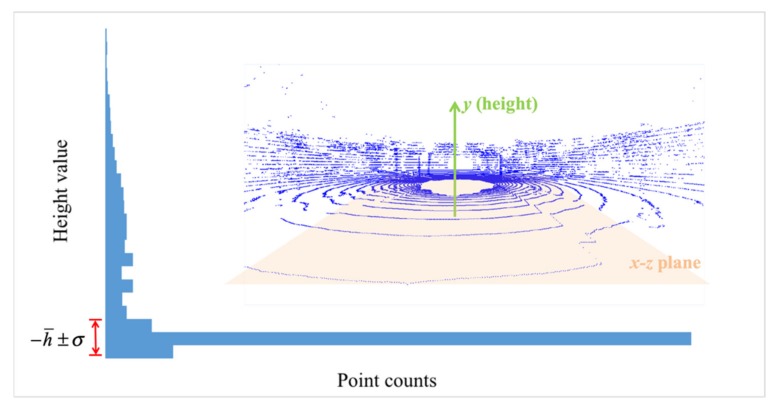
Height distribution of a frame of point cloud.

**Figure 3 sensors-20-02309-f003:**
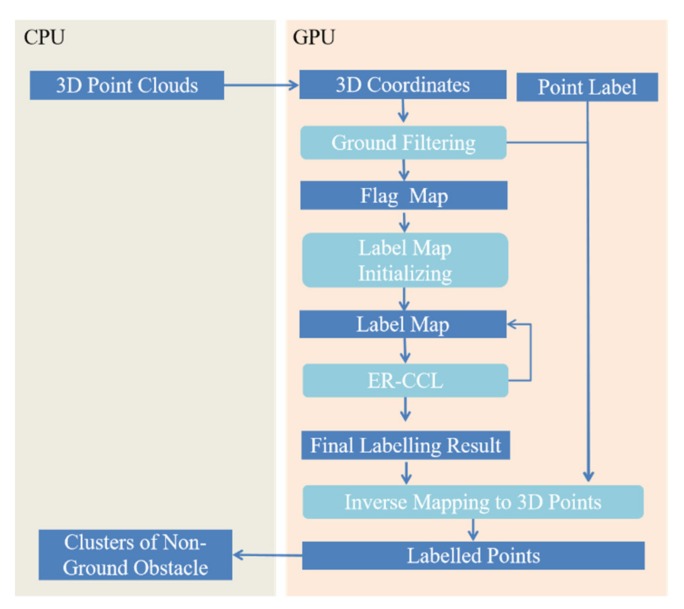
GPU accelerated framework for obstacle labeling in 3D point clouds.

**Figure 4 sensors-20-02309-f004:**
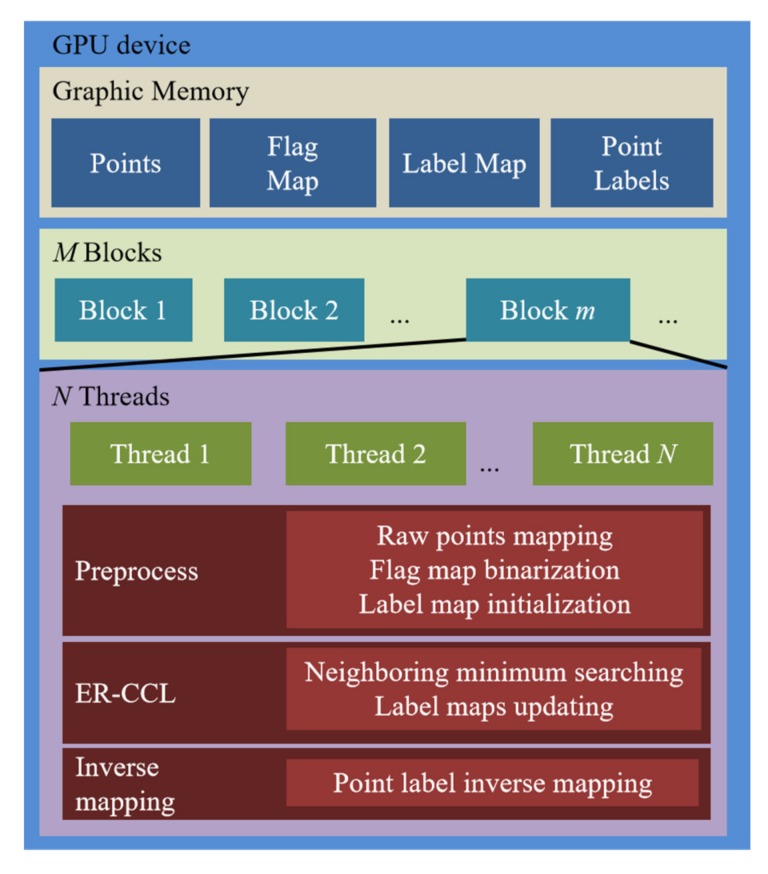
Memory and thread allocation in GPU device.

**Figure 5 sensors-20-02309-f005:**
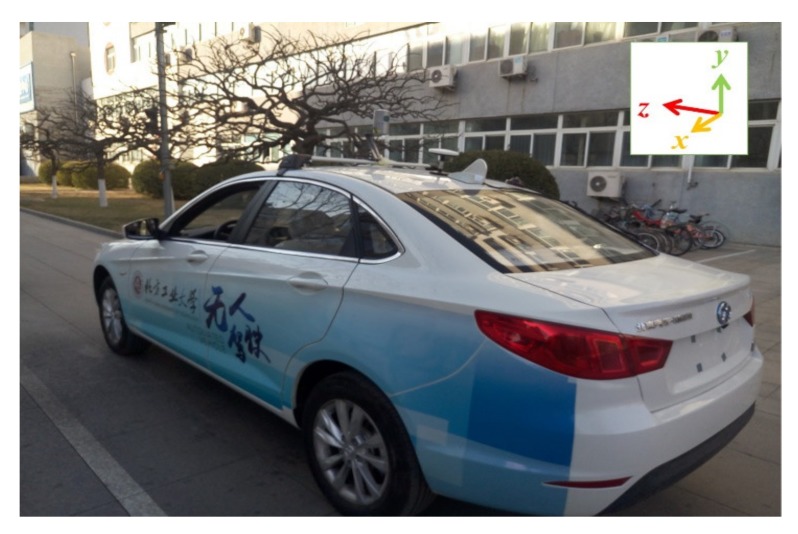
Unmanned vehicle carried with LiDAR sensor.

**Figure 6 sensors-20-02309-f006:**
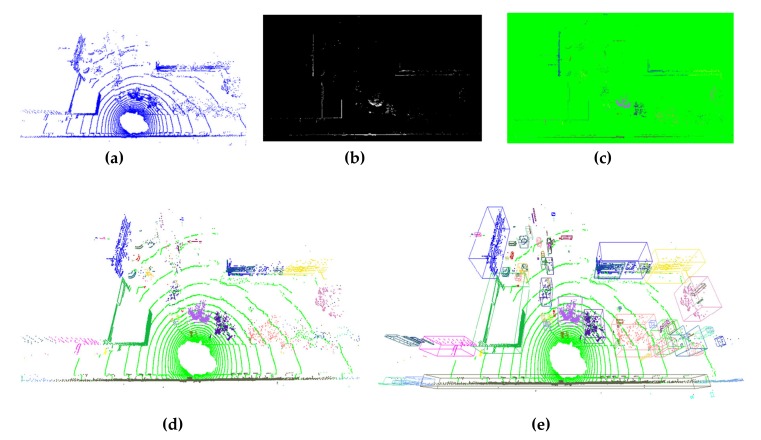
Experimental results obtained by the proposed 3D obstacle labeling method in LiDAR point clouds: (**a**) raw 3D points in the valid range; (**b**) generated obstacle flag map; (**c**) ER-CCL algorithm result on the label map; (**d**) inverse mapping result of 3D obstacle labeling; (**e**) obstacle labeling result with bounding boxes.

**Figure 7 sensors-20-02309-f007:**
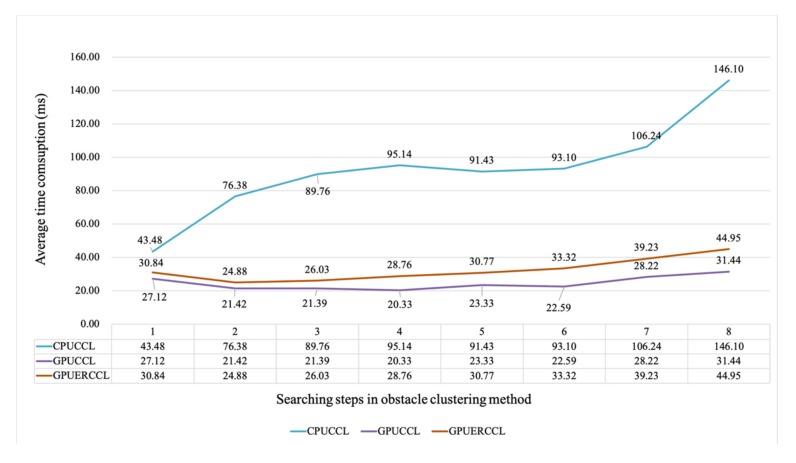
Speed performance with different numbers of search ranges.

**Figure 8 sensors-20-02309-f008:**
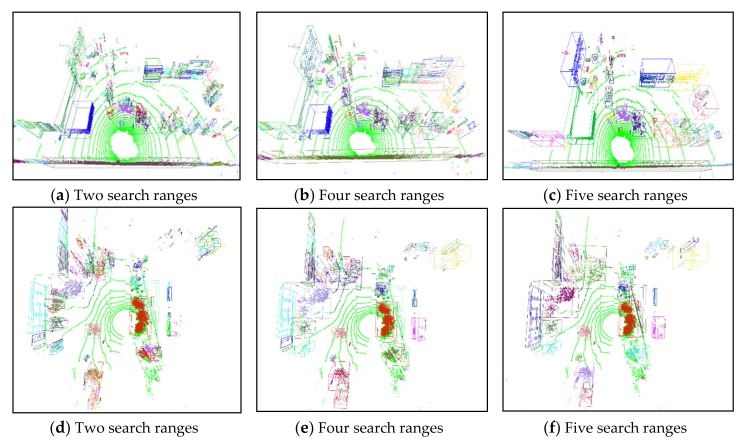
Obstacle clustering results obtained with different numbers of search ranges on different datasets: (**a**–**c**) clustering results of our collected dataset; (**d**–**f**) clustering results of NCLT dataset.

**Figure 9 sensors-20-02309-f009:**
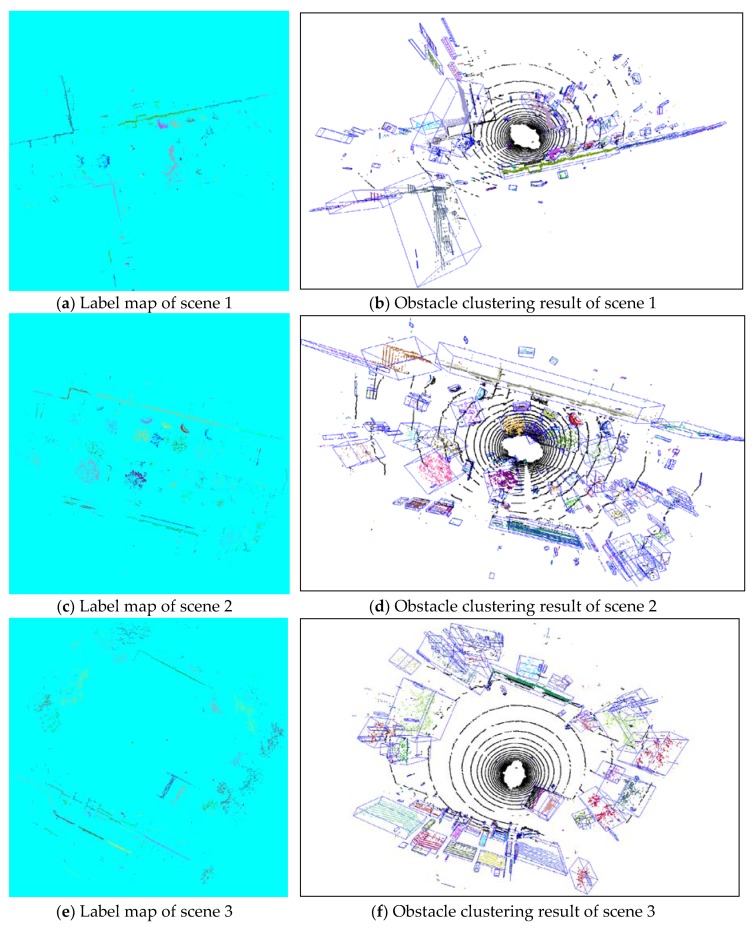
Obstacle clustering results in different scenes: (**a**) clustering result in label map in scene 1 (T junction); (**b**) obstacle clustering result in scene 1 (T junction); (**c**) clustering result in label map in scene 2 (road); (**d**) obstacle clustering result in scene 2 (road); (**e**) clustering result in label map in scene 3 (square); (**f**) obstacle clustering result in scene 3 (square); (**g**) clustering result in label map in scene 4 (multi-trees); (**h**) obstacle clustering result in scene 4 (multi-trees); (**i**) clustering result in label map in scene 5 (multi-person); (**j**) obstacle clustering result in scene 5 (multi-person); (**k**) clustering result in label map in scene 6 (crossroad); (**l**) obstacle clustering result in scene 6 (crossroad).

**Figure 10 sensors-20-02309-f010:**
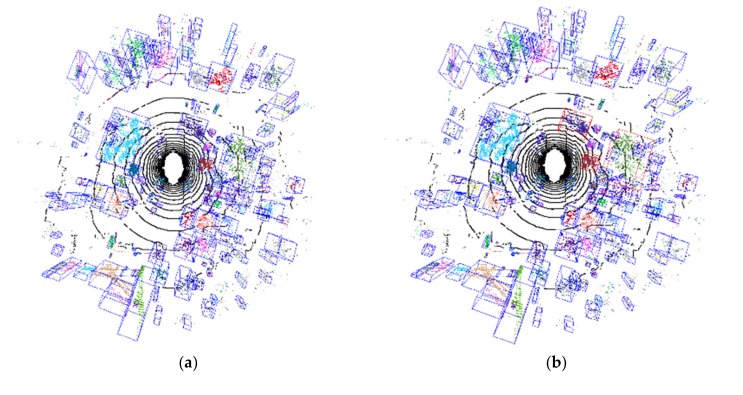
Obstacle clustering results in *x*–*z* horizontal plane obtained using three algorithms: (**a**) ER-CCL; (**c**) connected component analysis; (**e**) clustering in hash table; (**b**,**d**,**f**) Manually marked error clustering parts.

**Figure 11 sensors-20-02309-f011:**
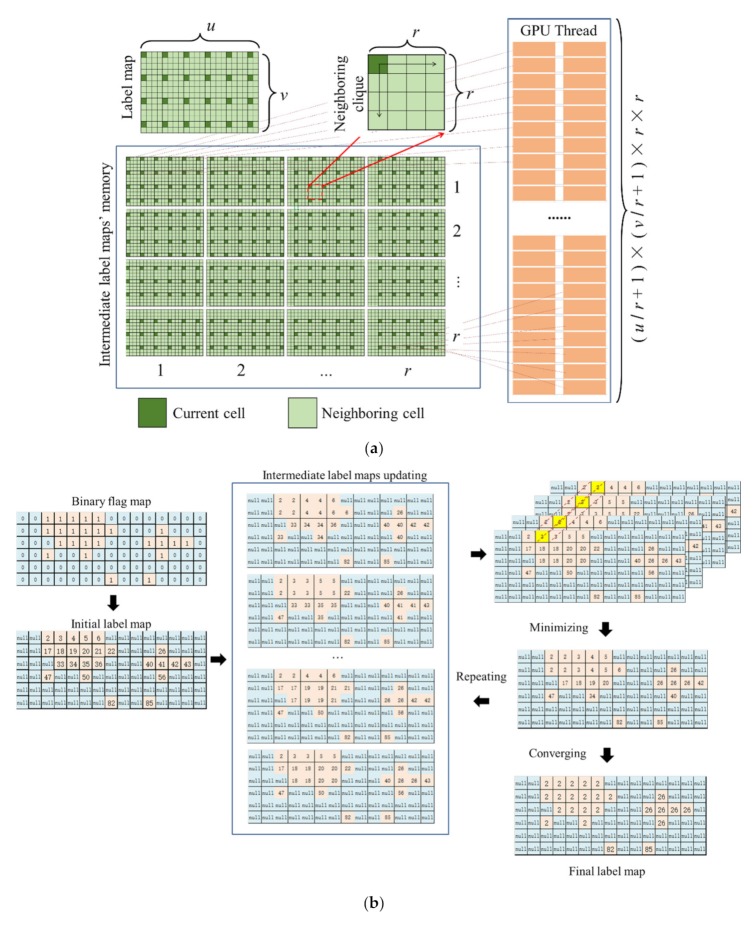
Proposed data independency solution of labeling process: (**a**) intermediate label map updating and thread allocating; (**b**) intermediate label maps merging process with independent data.

**Figure 12 sensors-20-02309-f012:**
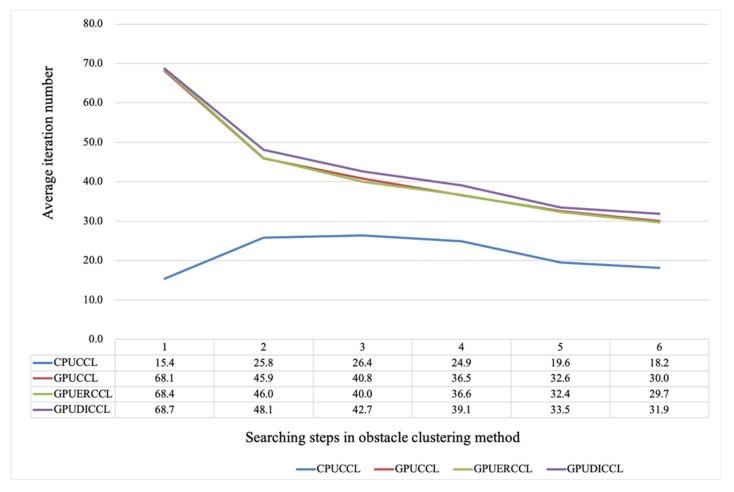
Iteration times under different searching ranges for different CLL implementation methods.

**Table 1 sensors-20-02309-t001:** Detail information of clustering results in different scenes

Scene No.	Cluster Count	Iteration Count	Time (ms)
Scene 1 (T junction)	301	50	39.85
Scene 2 (road)	528	79	52.16
Scene 3 (square)	493	39	40.14
Scene 4 (multi-trees)	682	46	46.58
Scene 5 (multi-persons)	474	46	40.02
Scene 6 (crossroad)	497	30	28.89

**Table 2 sensors-20-02309-t002:** Obstacle clustering results obtained using different algorithms

	Obstacle Count	Error Data Count	Clustering Accuracy	Time (s)
Elevation-reference CCL (our)	170	3	98.2%	0.02
Connected component analysis [33]	185	8	95.7%	0.04
Clustering in hash table [28]	130	6	95.4%	0.10

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
