# Peer review of "A Fast Spatial Clustering Method for Sparse LiDAR Point Clouds Using GPU Programming"

_sensors, 2020, doi:10.3390/s20082309_

Round 1

Reviewer 1 Report

This article discussed a GPU-accelerated spatial clustering method. It is well written and structured. That is a good idea, but a lack of new and significance. Most of the methods described in the paper are similar to those already used in other papers. 

There is no consideration of obstacles located below the ground. Self-driving vehicles do not only avoid obstacles higher than the ground when driving on their own. Therefore, authors should present how to cluster obstacles lower than the ground in the spatial clustering method.

For parallel processing on the GPU, it is essential to remove the dependency of data required for computation. In this paper, it is expected that these dependencies are eliminated by performing the same operation repeatedly. The authors should suggest another way to eliminate data dependencies for GPU parallelism.

Some minor issues

  • The abbreviation should be written out in full or explained the first time you use it.
  • In math equations on pages 5 and 6, there are two symbols for an apostrophe.
  • What is 250x32x12 on page 9?

Reviewer 2 Report

The paper presents a simple algorithm for Lidar data clustering using a method that the authors call connected component labelling (CCL). 
The method is relatively simple and uses the geometric characteristic of the points in a point cloud. 

The choice of the authors is to implement an algorithm that is based on the elevation of a point in contrast to other state-of-the-art algorithms. This places the paper in a non-main stream direction of research. Though in the related works some approach using learning algorithms and neural networks are presented, I think that the authors should better emphasize their approach as a non-neural network, and justify their choice. Why this is better than a learning algorithm?

Section 3 is clear, the schematic representation in Fig.2 is also helpful (a typo is present in the image "lablling result"). The method is simple, though a clear pseudocode of the algorithm can be useful to help the reader go through the section and to help future users to reproduce the results of the algorithm, the one presented at line 277 pag. 8 is very unclear. I suggest the authors to use a standard way to write a pseudocode. This must be done before the paper to be published. 

In Esperiment and Analysis", section 4, a "searching step" appear that is not well described in the methodology, this should be better explained in Section 3 before getting to the results section. Does using more "searching step" improve the clustering results? why one should use a higher value rather than a lower one?

Line 367: "We statistic the clustering details" ---- statistic is not a verb, what do the authors mean here?

I have not understood why in Section 4.5 the authors say that they compared the CCL with other three methods. In my understanding (also from Table 2) there is CCL (authors'), [36] and [37], total of three algorithm. Meaning that the CLL is compared against two additional methods. Also at line 401, they write "the four methods" and at line 408 "other three methods", am I missing something?

Minor issues: 

KNN abbreviation is used but not defined, though well known I guess k-nearest neighbors is worth to mention

English style should be revised as a many typos and unclear sentences are present in the paper, I point out just one example:

"When a GPU thread executes the label search process, the thread is
hold on to access GPU global memory that store the neighbor cells in flag map and obstacle label map."     global memory --- storeS?

lines 375-376: "Even though some obstacle that a little far with vehicle exist" this is not clear 

Round 2

Reviewer 1 Report

I appreciate the clarifications and improvements introduced by the authors in the revised manuscript.

Reviewer 2 Report

I think the paper is improved since last time. Now it can be published after a review to fix English style in the paper.